# System network analysis of *Rosmarinus officinalis* transcriptome and metabolome—Key genes in biosynthesis of secondary metabolites

Ali Moghadam ⓘ *, Eisa Foroozan, Ahmad Tahmasebi, Mohammad Sadegh Taghizadeh, Mohammad Bolhassani, Morteza Jafari

Institute of Biotechnology, Shiraz University, Shiraz, Iran

* ali.moghadam@shirazu.ac.ir

**Data Availability Statement:** All relevant data are within the paper and its Supporting information files.

## Abstract

Medicinal plants contain valuable compounds that have attracted worldwide interest for their use in the production of natural drugs. The presence of compounds such as rosmarinic acid, carnosic acid, and carnosol in *Rosmarinus officinalis* has made it a plant with unique therapeutic effects. The identification and regulation of the biosynthetic pathways and genes will enable the large-scale production of these compounds. Hence, we studied the correlation between the genes involved in biosynthesis of the secondary metabolites in *R. officinalis* using proteomics and metabolomics data by WGCNA. We identified three modules as having the highest potential for the metabolite engineering. Moreover, the hub genes highly connected to particular modules, TFs, PKs, and transporters were identified. The TFs of MYB, C3H, HB, and C2H2 were the most likely candidates associated with the target metabolic pathways. The results indicated that the hub genes including *Copalyl diphosphate synthase* (*CDS*), *Phenylalanine ammonia lyase* (*PAL*), *Cineole synthase* (*CIN*), *Rosmarinic acid synthase* (*RAS*), *Tyrosine aminotransferase* (*TAT*), *Cinnamate 4-hydroxylase* (*C4H*), and *MYB58* are responsible for biosynthesis of important secondary metabolites. Thus, we confirmed these results using qRT-PCR after treating *R. officinalis* seedlings with methyl jasmonate. These candidate genes may be employed for genetic and metabolic engineering research to increase *R. officinalis* metabolite production.

## Introduction

Global interest in the use of medicinal plants is growing and their beneficial effects on the production of drugs are being identified. It is estimated that only 10% of the 250,000 plant species important to human health have been studied [1]. On the other hand, essential oils are in high demand globally with a compound annual growth rate of 9.3%, which is expected to reach 16 million dollars by 2026 [2]. It is therefore necessary for researchers to find a way to increase the production of essential oils and derived compounds.

**Funding:** The author(s) received no specific funding for this work.

**Competing interests:** The authors have declared that no competing interests exist.

Due to its flavoring and therapeutic properties, *Rosmarinus officinalis* is the most commonly used medicinal herb among spices in food, pharmaceuticals, and cosmetics [3]. Previous studies have shown that *R. officinalis* has showed very high therapeutic effects due to the presence of bioactive compounds such as 1,8-cineol (15–55%), verbenone (2.2–11.1%), borneol (1.5–5.0%), camphor (5–31%), limonene (1.5–5.0%), pinene (2.0–9.0%), pinene (9–26%), caryophyllene (1.8–5.1%), camphene (2.5–12.0%), and myrcene (0.9–4.5%) [4, 5]. These metabolites have been applied for the prevention and treatment of various diseases such as colds, rheumatoid arthritis, and musculoskeletal discomforts [4, 5]. In addition, this plant presents a wide range of biological properties including antibacterial [6], antitumor [7], antidiabetic [8], antioxidant [9], and anti-inflammatory [10]. Accordingly, the rosemary extract was approved as food additives by EU Food Standards Agency (E392) [4]. Generally, medicinal plants are a renewable and almost unlimited source for new and complex chemical compounds [11–13]. According to these properties, research on this plant will be very interesting.

Although the secondary metabolites show diverse functions in the plants, including regulating growth [14], interacting with the environment [15], and ameliorating abiotic and biotic stresses [16, 17], the structure and metabolic pathways of secondary metabolites are still unknown. It is therefore crucial to identify key genes involved in these pathways in order to expand large-scale production and to maximize its potential in the fields of biotechnology, horticulture, and pharmacy. In recent years, the approach of gene co-expression analysis has become a powerful tool for predicting the function and role of genes. This vision provide a deeper understanding of metabolic pathways in plants and increase the potential for genome-scale pathways and systems metabolic engineering to produce more bioactive compounds [18]. In gene co-expression study, those genes that have a similar expression pattern among different samples are identified [19]. Genes involved in the biosynthesis of specific metabolites have a common regulatory network and most of these genes have closely related to each other (modules) in the co-expression network, which facilitate their identification [20]. However, the aim of this study is the identification of transcription factors, protein kinases, transporters, and hub genes involved in the biosynthesis pathway of rosmarinic acid using co-expression analysis and then evaluating the expression profile of hub genes in methyl jasmonate (MeJA)-treated *R. officinalis* seedlings.

## Materials and methods

### Data collection and processing

Metabolomics and transcriptomics data of *R. officinalis* were retrieved from the Plant/Eukaryotic and Microbial Systems Resource database (http://metnetweb.gdcb.iastate.edu/P-MR/) and Medicinal Plant Genomics Resource database (http://medicinalplantgenomics.msu.edu-/), respectively. First, the transcript expression level and the quantity of secondary metabolites within the tissues of transcriptomic and metabolomic data were classified correspondently. In addition, transcripts with low expression values observed in the transcriptomic data were also filtered out using the genefilter package based on the variance filtering through varFilter function. Then transcriptomics data is related to 32402 genes in 15 different samples i.e., young and mature leaf, secondary stem, root, callus, and flower. Transcriptomics and metabolomics data were integrated to identify the modules correlated with the biosynthesis of secondary metabolites by calculating module eigengenes (ME) and estimating Pearson's correlations between the MEs and selected metabolites to perform a comparative analysis using R Package Weighted Gene Co-expression Network Analysis (WGCNA). In this study, the expression value of Fragments Per Kilobase of exon per Million reads

(FPKM) and liquid chromatography/time-of-flight/mass spectrometry (LC/TOF/MS) of the metabolome profiles were used for the transcriptome and proteome data, respectively.

## Co-expression analysis

All samples were used to construct the co-expression modules related to the biosynthesis pathway of secondary metabolites. The co-expression network was constructed using the WGCNA [21]. The gene clusters were analyzed in the transcriptome datasets by measuring Pearson correlation coefficient using the power adjacency function [22]. Topological Overlap Matrix (TOM) and a hierarchical cluster tree were set up according to setting soft thresholding power β = 16 and transforming the Pearson correlation matrix into a weighted matrix. Therefore, a appreciate power value was measured to construct the network, which obtained by measuring the scale free fit index in a group of powers with a minimum of 1 to maximum of 30. The identification of gene co-expression modules was conducted by the blockwiseModules function and a dynamics tree cut algorithm. Finally, the correlation of the modules with the major secondary metabolites was investigated.

## Gene ontology and pathway analysis

The modules were obtained by BLASTX of transcript data in Swissport database, followed by submitted to the STRING database (https://string-db.org/) [23] to perform functional enrichment analysis. Afterwards, the Kyoto Encyclopedia of Genes and Genomes (KEGG) pathway enrichment analysis was performed using by the DAVID web tool (http://david.abcc.ncifcrf.gov/) with an FDR ≤ 0.05. Transcription factors (TFs) and protein kinases (PKs) were identified by BLASTX of transcript data in the iTAK database (http://bioinfo.bti.cornell.edu/cgibin/itak/index.cgi) [24]. In addition, the TCDB database (http://www.tcdb.org/) [25] was used to predict the transporters.

## Identification of hub genes

The network was visualized in Cytoscape software in the cytohubba plug-in. After calculating the connectivity scores using the Maximal Clique Centrality (MCC) function, the genes with the highest connectivity score in a module were considered as hub genes. The number of hub genes were selected based on the number of genes in each module.

## Plant selection and treatment

Many plant species treated with MeJA are able to produce compounds such as alkaloids, terpenoids, coumarins, and phenolic compounds [26]. On the other hand, the accumulation of rosmarinic acid in cell culture of MeJA-treated *Mentha balsamea* has been proven [27]. Recent studies have found that MeJA stimulates the synthesis of active compounds in rosemary suspension cells over a broad concentration range using concentration-dependent differential expression patterns. MeJA also reduced the peroxidative damage to the rosemary suspension cells over a broad concentration range using these same techniques [28, 29]. Therefore, MeJA was selected to elicit the expression of genes involved in the biosynthesis pathway of secondary metabolites in *R. officinalis*.

The seedlings were cultivated under optimum conditions in pot and then treated with MeJA in the concentration of 100 mM for 12 and 24 h in three biological replicates [30, 31]. After applying treatment, the 15 leaves were collected for each sample and frizzed in liquid nitrogen, followed by transferred to -80 ˚C for more analysis. Finally, we used just 100 mg leaf powder for RNA extraction for each sample. RNA extraction was carried out using the

**Table 1. The sequences of specific primers used to the Real-time PCR.**

| Gene | Sequence (5′ to 3′) |
|---|---|
| *Copalyl diphosphate synthase* (*CPS*) | F: GTTGGTGAAGTTAGTGCT |
| | R: GTCTCATCATCGTGGTAG |
| *Phenylalanine amonia-lyase* (*PAL*) | F: CTGGTCACTGCCTTCTAA |
| | R: CTGAGGTAGGGTATGGTC |
| *MYB 4* | F: TGGTAAGTGGTTGAGAATC |
| | R: AGACAGTGATGAGTAGCA |
| *Alpha tubulin* (*αTUB*) | F: GGAAATGCTTGCTGGGAG |
| | R: GAAGAAGGTGTTGAAGGC |
| *Cinenol synthase* (*CIN1*) | F: TGTATGACGAGTGATGGATGAG |
| | R: ACTTGCGTGGATGCTATCTT |
| *Rosmarinic acid synthase* (*RAS*) | F: ATGAAGATCGATATCACAGAC |
| | R: GTAGAAATGCACGCTGAG |
| *Tyrosine aminotransferase* (*TAT*) | F: TGCTTCCACGCTAGTAAT |
| | R: GATTGCCTCTCTGGTTTG |
| *Cinnamate 4-hydroxylase* (*C4H*) | F: TCGTGTTCGACATCTTCAC |
| | R: TACTGCTGCACCACCTTG |

RNX-PlusTM kit of Sinagene Co. (S-1020-1) according to the manufacturer's instructions. The cDNA synthesis kit of Thermo Fisher Scientific Co. (K1622) was used to synthesize the first strand cDNA.

## Gene expression analysis using qRT-PCR

Gene expression analysis was performed using Real-time PCR (Bio-Rad, Hercules, CA) using specific primers of hub genes (Table 1).

All primers were designed by Allele ID software. The diluted cDNA samples used as the template for qRT-PCR. The achieved data were analyzed using a thermocycler Line GeneK with the software Line GeneK Fluorescent Quantitative Detection system (BIOER Technology, Hangzhou, China). The mean of Ct values of the housekeeping gene of alpha tubulin was used as the internal control. The relative expression level was calculated using ΔΔCt method.

## Statistical analysis

Three technical replicates were applied for each sample in qRT-PCR. The data were analyzed using the SAS 9.4 software (SAS Institute Inc., USA) at a *P-value* of $\leq 0.05$. A mean comparison analysis was performed using the Tukey test. The graphs were plotted using GraphPad prism software v8 (GraphPad Software Inc., San Diego, CA).

## Results

## Evaluation of co-expression modules

This study presents the data collection and analysis process to identify some central genes involved in the biosynthesis of secondary metabolites in *R. officinalis*. The correlation among differentially expressed genes (DEGs) was evaluated using WGCNA R-package to identify these genes. The hierarchical clustering tree was produced with a power value of 16 and a scale independence of up to 0.9 through network topology analysis to determine the average degree of connectivity and the independence within co-expression modules (Fig 1).

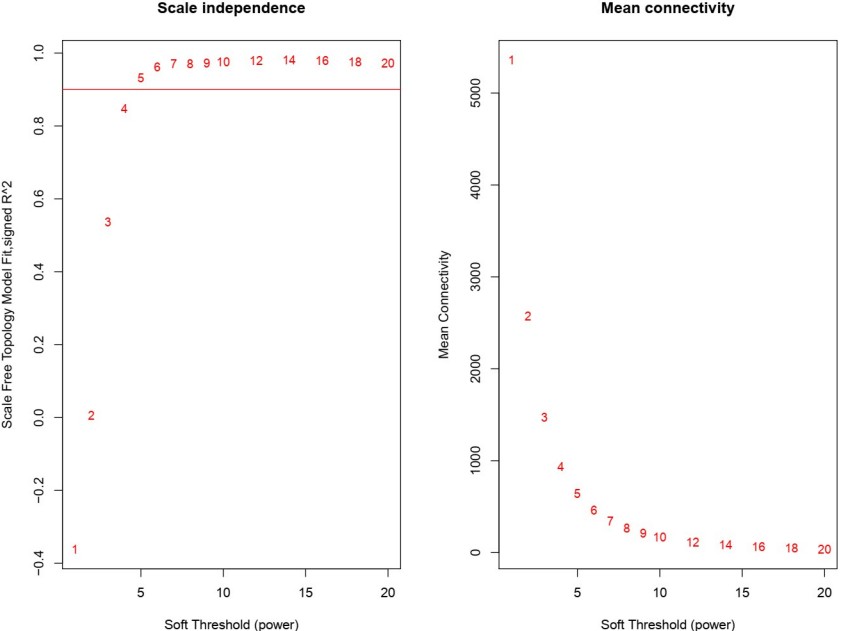

**Fig 1. Analysis of soft-thresholding power in the WGCNA.** The left picture displays determination of scale-free fit index (y-axis) for various soft-thresholding powers. The right picture shows determination of mean connectivity (degree, y-axis) for various soft-thresholding powers (x-axis).

Based on the eigengenes, an analysis of cluster for the modules of *R. officinalis* was performed. Results showed that modules were divided into 28 clades. Closed modules were combined by setting up the height of branch merge cut by 0.25 and showed in distinct colors (Fig 2a). Totally 24 co-expression modules were identified based on the dynamic tree cutting algorithm (Fig 2b). The modules related to important secondary metabolites were detected by the correlation between secondary metabolites and co-expression modules (Fig 2c).

Accordingly, the three major modules including brown4, green, and yellow, each containing 5567, 880, and 818 genes, respectively, showed higher levels of correlation with desirable secondary metabolites. These modules with the greatest potential for metabolic engineering were selected for further investigations.

## Related modules to secondary metabolites

According to our results, secondary metabolites have a significant correlation with co-expression modules, therefore we identified the modules with the most compounds. Intensive research has been conducted on rosemary and its components carnosol, carnosic acid, ursolic acid, rosmarinic acid, and caffeic acid over the past decade [32]. The results showed that the selected modules have the highest correlation with carnosol, carcinoic acid, and ursolic acid. Results indicate that rosmarinic acid and genkwanin, respectively, exhibited low and high correlations with all three modules, whereas genkwanin exhibited a remarkable correlation with brown4 module.

## Gene ontology and KEGG analysis

Gene Ontology (GO) and KEGG enrichment analysis were performed on the selected modules (Tables 2 & 3).

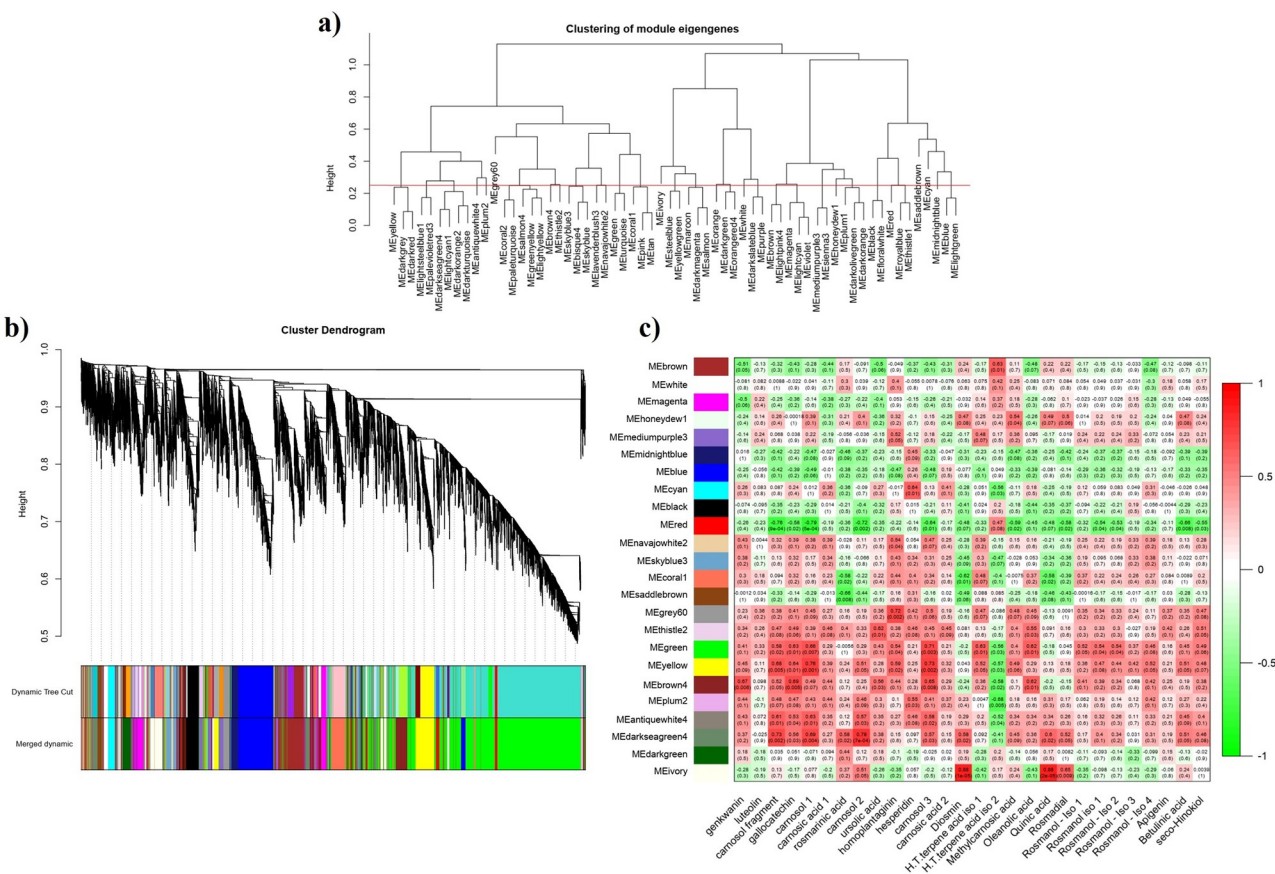

**Fig 2. The gene network analysis of the biosynthesis pathway of secondary metabolites in *R. officinalis*.** a) Clustering dendrogram of the modules, whereas the horizontal line showing merge cut height of 0.25. b) Co-expression modules constructed. Dendrograms produced by average linkage of hierarchical clustering of genes, which is based on a topological overlap matrix (TOM). The modules were assigned colors as indicated in the horizontal bar beneath the dendrogram. c) Heat map of correlation between eigengene modules and secondary metabolites. Each row shows a module eigengene, each column shows a secondary metabolite. Each cell contains the corresponding correlation and *P-value*. The correlation between modules and secondary metabolites in the cells has a ratio ranged from -1 to +1. The more this ratio closer to +1, the color changes from green to red.

**Table 2. GO enrichment analysis of genes in the selected modules.**

| Module | GO No. | Term | Count | FDR |
|---|---|---|---|---|
| Brown4 | GO:0009651 | Response to salt stress | 24 | 0.0092 |
| | GO:0009408 | Response to heat | 12 | 0.0350 |
| Green | GO:0055114 | Oxidation-reduction process | 295 | 2.59E-13 |
| | GO:0046686 | Response to cadmium ion | 81 | 8.18E-04 |
| | GO:0006810 | Transport | 78 | 0.0064 |
| | GO:0009409 | Response to cold | 71 | 0.0022 |
| | GO:0015979 | Photosynthesis | 66 | 1.25E-17 |
| Yellow | GO:0055114 | Oxidation-reduction process | 56 | 0.0022 |
| | GO:0006468 | Protein phosphorylation | 37 | 0.0354 |
| | GO:0006633 | Fatty acid biosynthetic process | 18 | 4.41E-06 |
| | GO:0080167 | Response to karrikin | 16 | 5.45E-05 |
| | GO:0007169 | Transmembrane receptor protein | 16 | 5.45E-05 |
| | | Tyrosine kinase signaling pathway | | |

**Table 3. KEGG pathway enrichment analysis of genes in the selected modules.**

| Module | KEGG No. | Pathway | Count | FDR |
|--------|----------|---------|-------|-----|
| Brown4 | - | - | - | - |
| Green | ath01100 | Metabolic pathways | 378 | 5.16E-14 |
| | ath01110 | Biosynthesis of secondary metabolites | 237 | 7.33E-13 |
| | ath01130 | Biosynthesis of antibiotics | 101 | 9.50E-05 |
| | ath01200 | Carbon metabolism | 72 | 5.22E-06 |
| | ath00710 | Carbon fixation in photosynthetic organisms | 32 | 8.32E-08 |
| Yellow | ath01110 | Biosynthesis of secondary metabolites | 45 | 1.67E-05 |
| | ath01212 | Fatty acid metabolism | 10 | 2.80E-04 |
| | ath00061 | Fatty acid biosynthesis | 8 | 2.80E-04 |
| | ath00073 | Cutin, suberine and wax biosynthesis | 7 | 2.80E-04 |
| | ath00062 | Fatty acid elongation | 6 | 0.0092 |

Functional enrichment showed that genes in the module brown4 were mainly enriched in response to salt stress (GO:0009651) and response to heat (GO:0009408). Genes in the module green were mainly enriched in oxidation-reduction process (GO:0055114), response to cadmium ion (GO:0046686), transport (GO:0006810), response to cold (GO:0009409), and photosynthesis (GO:0015979). Genes in the module yellow were mainly enriched in oxidation-reduction process (GO:0055114), protein phosphorylation (GO:0006468), fatty acid biosynthetic process (GO:0006633), response to karrikin (GO:0080167), and transmembrane receptor protein tyrosine kinase signaling pathway (GO:0007169).

## Identification of transcription factors

Since the TFs regulate the genes involved in the biosynthesis of secondary metabolites [33], the identification of TFs-related genes in the modules was performed by alignment of transcript sequences against the iTAK database. We identified 132 TFs in selected modules that are associated with metabolic pathways, among which MYB, Cysteine3Histidine (C3H), homebox (HB), and Cys2-His2 (C2H2) are the most likely candidates (Table 4).

## Identification of protein kinases

Recognition of PKs as regulators of critical signaling in response to environmental stresses will be very effective in improving the biosynthesis pathway of secondary metabolites [34]. In the selected modules, 220 genes encoding PKs were identified and classified into (PKA, PKG, PKC (AGC)), calmodulin-dependent protein kinase (CAMK), cytokeratin 1 (CK1), (CDK, MAPK, GSK3, CLK kinases (CMGC)), receptor-likekinases (RLK)-Pelle, homologs of yeast sterile (STE), and tyrosine kinase-like (TKL) families (Table 5).

The CMGC family was the largest among those present in the brown4 module. The RLK-Pelle_DLSV subfamily of RLK-Pelle and TKL families were the only family that was commonly identified in all the selected modules (S1 Table). Based on the results, the green module showed the most variable PKs in the selected modules.

## Identification of transporters

According to the key role of transporters in transport of plant secondary metabolites [35], we examined 7,265 unigenes in selected modules that were identified in the TCDB database. The TCDB classified transporters into 7 systems (Fig 3, S2 Table).

**Table 4. TF families in the selected modules.**

| Brown4 | | Green | | Yellow | |
|---|---|---|---|---|---|
| B3 | 2 | bHLH | 28 | MYB | 8 |
| GARP-G2-like | 2 | WRKY | 21 | HB-HD-ZIP | 6 |
| MADS-M-type | 2 | NAC | 17 | WRKY | 5 |
| Trihelix | 2 | C2H2 | 14 | SBP | 5 |
| bZIP | 2 | AP2/ERF-ERF | 13 | OFP | 4 |
| GRAS | 2 | bZIP | 13 | bHLH | 4 |
| MYB | 2 | MYB-related | 12 | bZIP | 3 |
| TUB | 2 | C2C2-Dof | 11 | C2C2-YABBY | 3 |
| C2H2 | 2 | GARP-G2-like | 11 | MYB-related | 3 |
| C3H | 2 | MYB | 10 | GARP-G2-like | 2 |
| DDT | 1 | zf-HD | 9 | GRAS | 2 |
| MADS-MIKC | 1 | LOB | 9 | C2C2-GATA | 2 |
| TCP | 1 | GRAS | 9 | HB-other | 2 |
| bHLH | 1 | TCP | 8 | B3-ARF | 2 |
| HB-HD-ZIP | 1 | C3H | 7 | C2H2 | 2 |
| NAC | 1 | C2C2-GATA | 6 | HB-WOX | 2 |
| WRKY | 1 | Trihelix | 5 | NAC | 2 |
| HSF | 1 | HSF | 4 | AP2/ERF-RAV | 1 |
| SBP | 1 | OFP | 4 | NF-YA | 1 |
| | | C2C2-YABBY | 4 | Trihelix | 1 |
| | | C2C2-CO-like | 3 | AP2/ERF-AP2 | 1 |
| | | SRS | 3 | LOB | 1 |
| | | AP2/ERF-AP2 | 3 | AP2/ERF-ERF | 1 |
| | | PLATZ | 3 | PLATZ | 1 |
| | | B3 | 3 | zf-HD | 1 |
| | | B3-ARF | 3 | HB-BELL | 1 |
| | | HB-HD-ZIP | 3 | C3H | 1 |
| | | E2F-DP | 3 | SRS | 1 |
| | | MADS-M-type | 3 | | |
| | | DBB | 2 | | |
| | | HB-BELL | 2 | | |
| | | TUB | 2 | | |
| | | HB-other | 2 | | |
| | | CPP | 1 | | |
| | | EIL | 1 | | |
| | | GRF | 1 | | |
| | | CSD | 1 | | |
| | | GARP-ARR-B | 1 | | |
| | | HB-WOX | 1 | | |
| | | LIM | 1 | | |
| | | RWP-RK | 1 | | |
| | | C2C2-LSD | 1 | | |
| | | DBP | 1 | | |
| | | GeBP | 1 | | |
| | | MADS-MIKC | 1 | | |
| | | NF-YA | 1 | | |
| | | S1Fa-like | 1 | | |

(*Continued*)

**Table 4.** (Continued）

| Brown4 | | Green | | Yellow | |
|---|---|---|---|---|---|
| | | BES1 | 1 | | |
| | | CAMTA | 1 | | |
| | | NF-YB | 1 | | |
| | | SBP | 1 | | |
| | | VOZ | 1 | | |

MYB is a very large TF family that was detected in every selected module except green. Here, C3H has been detected in all selected modules. GAI-RGA- and -SCR (GRAS), SQUAMOSA promoter binding protein (SBP), Cys2Cys2 (C2C2), and homebox homeodomain leucine zipper (HB-HD-ZIP) play a major role in all modules, as can be seen in the results (Table 4). In addition, MCM1, AG, DEF and SRF (MADS) is an important family of proteins that was identified in all modules except for yellow. According to the results, the yellow module includes a higher number of TFs than the other modules.

This database contains 7 main systems with subsystems. The main systems are listed as follows: channels/pores, electrochemical potential-driven transporters, primary active transporters, group translocators, transmembrane electron carriers, accessory factors involved in transport, and incompletely characterized transport systems [36]. According to the results, 25.7% of transporters were classified in incompletely characterized transport systems, which means that many transporters of *R. officinalis* have potential for exploring their structures. Based on the results, most transporters including Voltage-gated Ion Channel (VIC) superfamily, Gap Junction-forming Innexin family, MCA family, nuclear pore complex (NPC) family, and membrane-limited channels, belonged to the channel/pore family, where the share of channel types is higher. The VIC superfamily was identified in 6.4%, 7.7% and 9.7% of transporters in the brown4, green and yellow modules, respectively. Electrochemical potential-driven transporters, which part of the Major Facilitator Superfamily (MFS) [25], were identified in all selected modules in 3.9%, 3.4% and 2.8% in the brown4, green and yellow modules, respectively. Primary active transporters were also identified in all selected modules as 0.2%, 0.7% and 0.2% in the brown4, green, and yellow modules, respectively (Fig 3). Study of the transporters shows that the green module was more active than the other selected modules in secondary metabolites transformation, as it provides a higher expression level of major transporters.

### Identification of hub genes

Hub genes were identified using Cytoscape software (Cytohubba plugin) and grouped by the MCC method (Fig 4).

**Table 5. PK families in the selected modules.**

| Brown4 | | Green | | Yellow | |
|---|---|---|---|---|---|
| TKL | 5 | RLK-Pelle | 101 | RLK-Pelle | 36 |
| CMGC | 4 | CAMK | 18 | AGC | 3 |
| CAMK | 3 | TKL | 13 | TKL | 2 |
| RLK-Pelle | 3 | AGC | 12 | Others | 1 |
| Plant-specific | 2 | CMGC | 6 | STE | 1 |
| CK1 | 1 | Others | 5 | | |
| | | Plant-specific | 2 | | |
| | | STE | 2 | | |

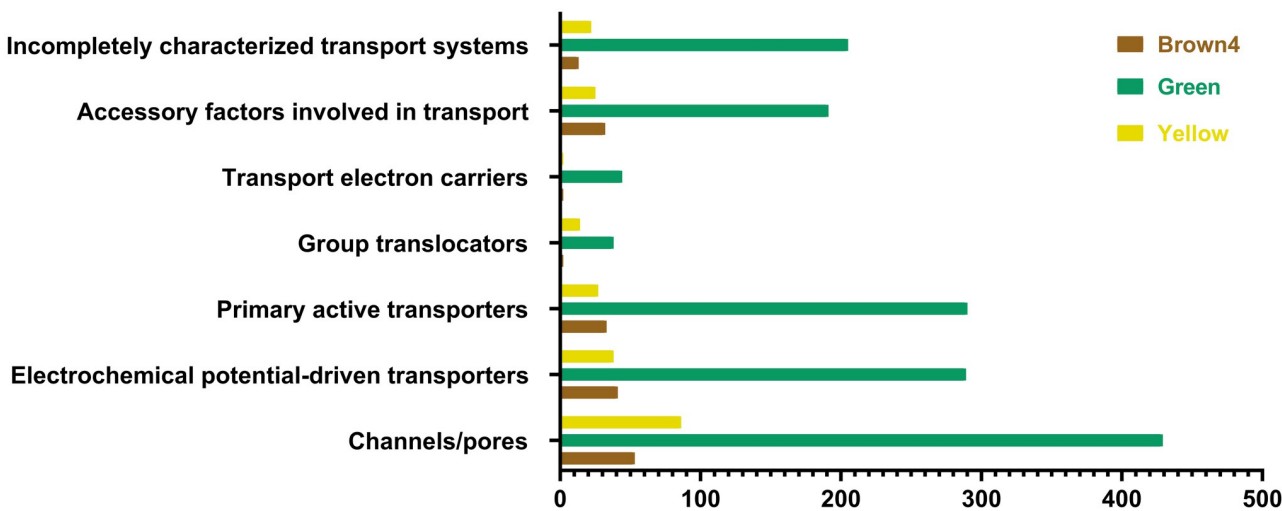

**Fig 3. Transporter systems in selected modules of *R. officinalis*.** Y-axis shows the transporter systems and x-axis shows the number of unigenes.

Among the 80 hub genes completely screened out in the current study, some genes with unknown functions may be worth exploring further. Following their strongest connections, the top 20 genes were selected as hubs for brown4 module and the 30 top genes were selected for both green and yellow modules. Among hub genes identified in the selected modules, *Cineole synthase* (*CIN1*), *Copalyl diphosphate synthase* (*CPS*), *Rosmarinic acid synthase* (*RAS*), and *Tyrosine aminotransferase* (*TAT*) were placed in yellow modules. In addition, *Phenylalanine ammonia-lyase 2* (*PAL*) and *Cinnamate 4-hydroxylase* (*C4H*) were placed in brown4 module, as well as *MYB58* (*MYB*) were placed in green module (Table 6). Seven hub genes were selected for the qRT-PCR and their expression profile was confirmed in MeJA-treated *R. officinalis* seedlings.

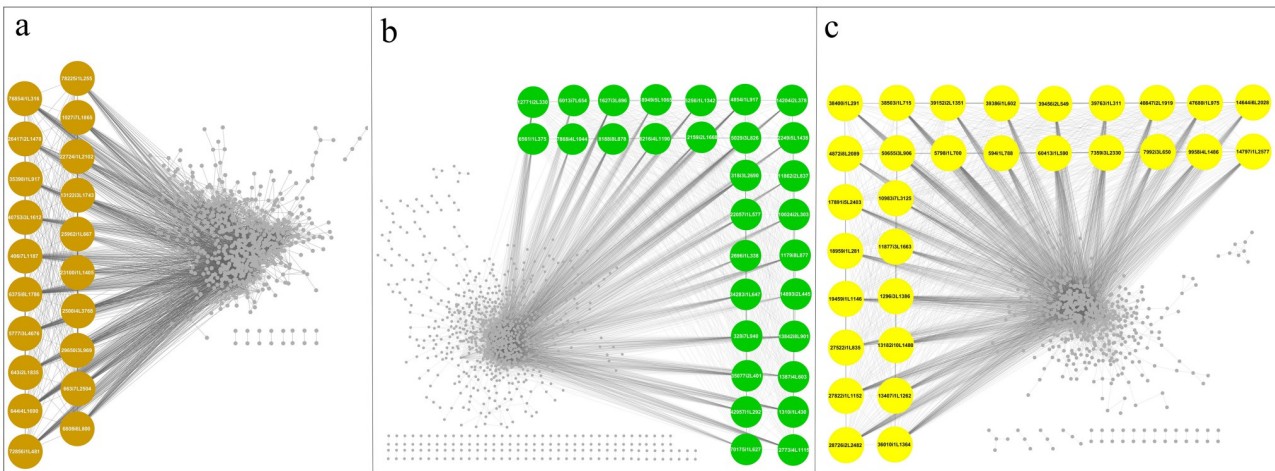

**Fig 4. Identification of hub genes involved in metabolite biosynthesis pathway of *R. officinalis*.** Diagrams show the identified hub genes in the brown4 (a), green (b), and yellow (c) modules. Nodes represent genes and brown, green, and yellow bigger nodes indicate the hub genes. The gray line connecting two nodes indicates the correlation between them.

**Table 6. The identified hub genes in correlation with rosmarinic and carnosic acid pathways.**

| Genes | Module | Correlation |
|---|---|---|
| *Cineole synthase* (*CIN1*) | Yellow | **74–89%** |
| *Copalyl diphosphate synthase* (*CPS*) | Yellow | **74–89%** |
| *Rosmarinic acid synthase* (*RAS*) | Yellow | **74–89%** |
| *Gernylgeranyl reductase* | Green | **72–86%** |
| *MYB 4* | Green | **72–86%** |
| *R2r3-myb58 transcription factor* | Green | **72–86%** |
| *Cytochrome P450* (*CYP76C4*) | Yellow | **74–86%** |
| *Prephenate aminotransferase* (*PAT*) | Brown4 | **72–89%** |
| *Tyrosine aminotransferase* (*TAT*) | Yellow | **74–86%** |
| *Gernylgeranyl diphosphate synthase* (*GGPS1*) | Yellow | **74–86%** |
| *Phenylalanine ammonia-lyase 2* (*PAL2*) | Brown4 | **72–89%** |
| *Cinnamate 4-hydroxylase* (*C4H*) | Brown4 | **72–89%** |
| *Phenylalanine ammonia-lyase 1* (*PAL1*) | Green | **72–85%** |
| *Hydroxycinnamoyl-CoA shikimate: quinate hydroxycinnamoyl transferase* (*HCT*) | Green | **72–85%** |
| *Cytochrome P450* (*CYP98A3*) | Brown4 | **72–89%** |
| *Limonene synthase* | Yellow | **74–86%** |
| *Chorismate synthase* | Brown4 | **72–89%** |

## qRT-PCR of hub genes

The relative expression analysis of seven hub genes, including *PAL*, *MYB*, *CPS*, *CIN1*, *TAT*, *C4H*, and *RAS*, showed that all of the genes increasingly expressed after MeJA treatment in comparison with control after 12 h. After 12 h of treatment, the highest and lowest of relative expression level were respectively related to *PAL* and *MYB* (Fig 5a & 5b). Accordingly, the expression level of the *PAL*, *CPS*, and *MYB* was about 7.6-, 5.7-, and 1.27-fold in comparison with the control after 12 h, respectively (Fig 5a–5c). After 24 h, the expression level of *CPS* and *PAL* decreased sharply by 0.46- and 0.5-fold, respectively, whereas *MYB* expression was increased by 3-fold (Fig 5a–5c). As the results showed, the *CIN1* significantly up-regulated at 12 and 24 h (4.3 and 6.9-fold in comparison with the control, respectively) (Fig 5d). In addition, *TAT* and *C4H* showed significantly up-regulated at 12 and 24 h (3.41 and 4.64-fold for *TAT* and 2.52 and 5.14-fold for *C4H*, respectively) (Fig 5e & 5f). Similarly, the relative expression of *RAS* increased in 12 and 24 h by 2.62 and 3.36-fold, respectively (Fig 5g).

Interestingly, the expression level of *CIN*, *MYB*, *RAS*, *TAT*, and *C4H* increased over the time. It can be suggested that the MeJA plays an elicitor role in more production of secondary metabolites related to these genes.

## Discussion

### Functional enrichment analysis of modules

There are a variety of secondary metabolites in *R. officinalis*, each of which is useful in treating disease [5]. Identifying the hub genes involved in the biosynthesis of each secondary metabolite is crucial for the development of pharmaceuticals and plant breeding programs, as well as industrial applications. In rosemary, the most important biological activities are associated with polyphenols such as carnosol and carnosic acid [37, 38]. Unigenes in brown4, green, and yellow were functionally classified into various categories by GO term enrichment analysis. In the biological process, GO terms of cellular process, transmembrane transport, metabolite

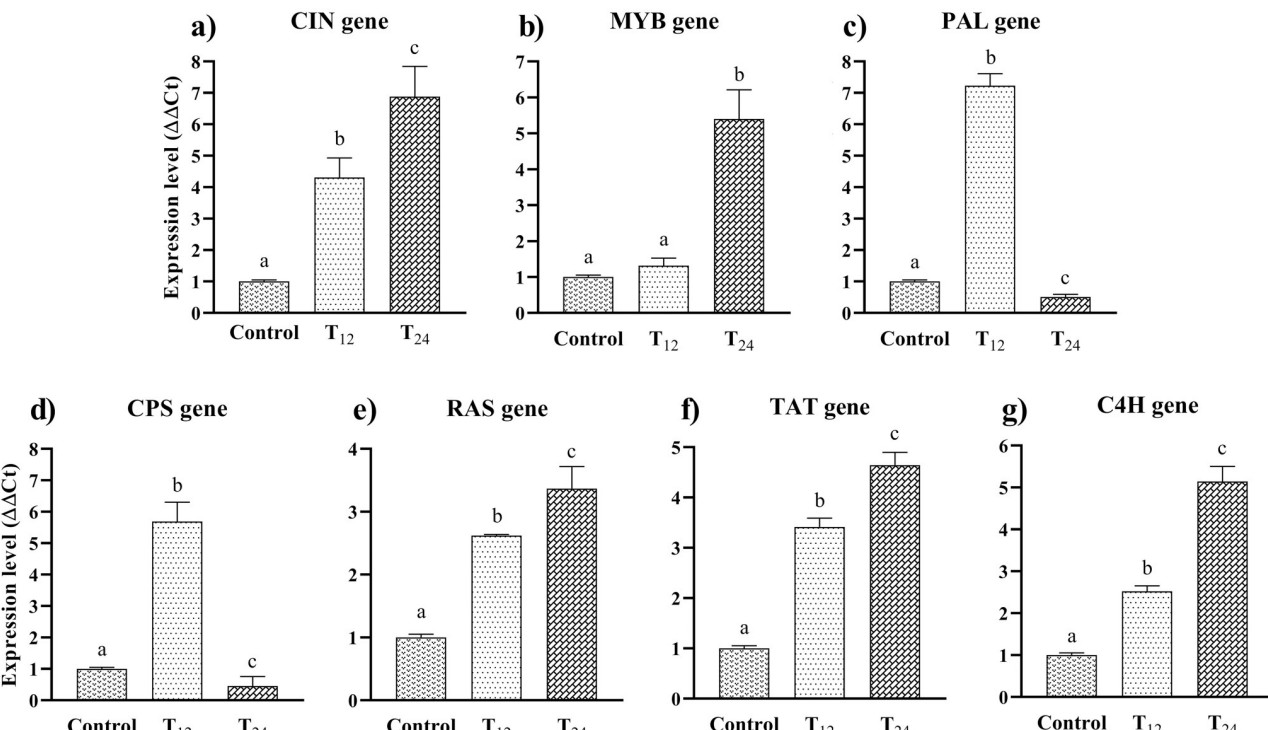

**Fig 5. Relative expression of hub genes under MeJA treatment using qRT-PCR.** The graphs show the expression level of the *PAL* (a), *MYB* (b), *CPS* (c), *CIN* (d), *TAT* (e), *C4H* (f), and *RAS* (g) after exposure times of 12 and 24 h in comparison with the control. Data were analyzed using the GraphPad prism at *P-value* of < 0.01 and expressed as mean ± SD (standard deviation).

process, and responses to metabolites and elements were more significant than other terms. Among the modules, the genes classified in the green module are most associated with the production of secondary metabolites (Table 2). Based on the KEGG enrichment analysis, biosynthesis of secondary metabolites was found as a noteworthy significant pathway for the green and yellow modules (Table 3).

In addition, results show that some of the co-expression genes are responsible for vital cellular functions. In brown4 and green modules, *PAL2* and *PAL1* were respectively identified as key genes in the rosmarinic acid biosynthesis pathway. Studies on *Satureja spicigera* showed that PAL is an essential enzyme in the biosynthesis of rosmarinic acid and phenolic compounds because of the expression of *PAL* and the total amount of rosmarinic acid is significantly correlated [39]. In addition, *TAT*, *RAS*, and *Cytochrome P450*, which identified in yellow module, are key genes in the rosmarinic acid biosynthesis pathway. Based on the presence of these genes, it is a key module in the pathway of rosmarinic acid biosynthesis [40]. The carnosic acid biosynthesis pathway starts with Granylgranyl diphosphate, which synthesized by Granylgranyl diphosphate synthase (GGPS), followed by copalyl diphosphate, synthesized by *CPS*, and finished by contribution of Cytochrome P450 [41]. According to identification of *GGPS*, *CPS* and *Cytochrome P450* in the yellow module, this module has a huge role in the carnosic acid biosynthesis.

## TFs, key regulatory proteins in metabolic pathways

In understanding how gene networks are regulated, it is important to understand how TFs control the biosynthesis of metabolites. Based on the identification of TFs in different

rosemary modules, 132 TFs, belonging to 34 TF families, are associated with metabolic pathways, but MYB (v-myb myeloblastosis viral oncogene homolog), C3H, HB, and C2H2 are the most likely ones (Table 4). Hence, MYB is a large TF family and was identified in all selected modules except green. The MYB TFs are the most abundant, regulating the biosynthesis of secondary metabolites [42]. In addition to regulating secondary metabolism, responses to hormones and environments, and cell differentiation and morphogenesis, MYB TFs also play a crucial role in resistance to drought and other abiotic stresses [43]. In *Arabidopsis* overexpressing *OsMYB3R-2*, the expression of *dehydration-responsive element-binding protein 2A*, *COR15a*, and *RCI2A* was significantly increased, leading to enhanced abiotic stress tolerance [44]. C3H were detected in all selected modules. C3H is zinc finger proteins involved in pre-mRNA processing, transcriptional regulation, and stress tolerance. C2H2 identified in all selected modules and C2C2 identified in green and yellow modules, both known as zinc finger TFs. In *Catharanthus roseus*, protein complexes containing C2H2 zinc fingers bind to the monoterpenoid indole alkaloid biosynthesis gene promoter and control the promotor activity [45]. In *Arabidopsis*, OBP2 designated TF, which is a C2C2-Dof (DNA-binding One Zinc Finger), showed an increase in the regulation of indole glucosinolates biosynthesis [45].

Studies on *C. roseus* under MeJA treatment showed that CrMYC2, an MYC family TF of bHLH, is contribution in the regulation of terpenoid indole alkaloids biosynthesis genes [28]. Studies showed that WRKYs could regulate the expression of genes that contribution in terpenes biosynthesis pathways through control the promoters of related genes [46]. Among bioactive compounds of *R. officinalis*, verbenone and camphor known as terpene biocompounds, in the other hand WRKY is identified in all selected modules. In addition, studies on *Cinnamomum camphora* showed that WRKY, MYB, AP2/ERF, bZIP and bHLH are regulate borneol biosynthesis pathway genes [47]. HB-HD-ZIP also plays a crucial role in all selected modules. Aliakbari and coworkers found that the expression of the protein group HB-HD-ZIP is effective in increasing secondary metabolites [48]. One of the protein families that play an important role in the modules is GRAS, which was identified in all modules. Homologs of GRAS proteins can be found in Arabidopsis, tomato, petunia, lily, rice and barley. GRAS family members are essential for physiological processes such as gibberellic acid (GA) signal transduction, stem cell maintenance, axillary meristem initiation, light signaling, photochromic signaling, male gametogenesis, and detoxification [49, 50]. According to the findings, this protein family plays a very important role in metabolism, especially under stressful conditions, and is thought to be involved in the production of secondary metabolites. In all plants, SBP protein family plays an important role in the process of flowering and reproduction, as well as regulating the ripening of fruits and ovarian function [51, 52], so that it found in all modules in this study. GA and anthocyanins are synthesized by this protein family, and they play a crucial role during abiotic stresses [53, 54]. MADS, found in all modules except yellow, is another important family of proteins. MADS boxes regulate the activity of topoisomerase PA, which is involved in DNA replication [55], temperature responses and flowering process regulation and also play a part in protein-protein interactions [56, 57].

Therefore, based on our results and previous researches, this protein family is likely caused by temperature-dependent environmental conditions in the regulation of secondary metabolites. Totally, TF analysis of *R. officinalis* enlighten that bHLH and WRKY are major protein families that are potentially contribute in synthesis pathways of biocompounds such as verbenone, camphor and borneol, also have noteworthy quantity in the selected modules.

## PKs involved in biosynthesis of secondary pathways

We have investigated potential PKs to find out how signal transduction occurs in rosemary. PKs play a pivotal role in regulating eukaryotic cell cycle and division [58]. From the iTAK database, 220 PKs were identified in 11 families. RLK-Pelle_DLSV (Receptor-Like Kinase/Pelle) subfamily is one of the PKs, which was identified in all three modules. According to the previous studies, this subfamily is involved in biotic and abiotic stresses [59, 60]. Leucine-rich repeats (LRRs) are a subfamily of RLK-Pelle identified in all modules. In a study, it has shown that LRR-RLKs play important roles in defense-related responses [61]. Receptor-like cytoplasmic kinase (RLCK) is another RLK subfamily that has 17 and 4 genes encoding in brown4 and yellow modules, respectively. The other hand, RLCKs are involved as mediators of cellular signaling in response to developmental and environmental signals [62]. In *Marchantia polymorpha*, an early-divergent land plant lineage, RLKs and SA pathway genes have been found [63]. In fact, a conserved biochemical defense response has been found in *M. polymorpha*, namely the role of PP-associated metabolites in mitigating pathogen infection [64]. LRR-RLKs have been shown to be involved in the cotton fiber development as well as cell wall biosynthesis in cotton. *RLK1* was induced in developing cotton fibers and was predicted to be involved in the secondary cell wall synthesis in cotton fiber [65].

In addition, tyrosine kinase like (TKL) is another group of PKs that was identified in all selected modules. In this family, biological roles in development, stress response and infection process were demonstrated using the gene silencing method [66]. AGC (protein kinase A/protein kinase G/protein kinase C-family) is another group of PKs identified in green and yellow modules, which has a direct effect on auto-phosphorylation [67]. Studies have shown that AGC protein kinase plays an important role in growth control in a regulatory cycle [68]. Some of the best-studied plant AGC kinases mediate auxin signaling and are thereby involved in the regulation of growth and morphogenesis. Furthermore, certain members are regulated by lipid-derived signals via the 3-phosphoinositide-dependent kinase 1 (PDK1) and the kinase target of rapamycin (TOR), similar to its animal counterparts [69]. $Ca^{2+}$/calmodulin-dependent protein kinase) CAMK (group contains a total of 21 encoding genes in brown4 and green modules. CAMK is directly and indirectly regulated by $Ca^{2+}$ and is involved in the response to hormone signaling, various biotic and abiotic stresses [70, 71]. It may be suggested that the mentioned gene families are part of the necessary enzymes for the production of secondary metabolites in *R. officinalis*.

Therefore, these gene families either can act as signal transmitters, like LRRs and RLCKs, which are the main building blocks of the metabolite production system, or participate in defense pathways, like TKL and RLK-Pelle, which are thought to be engaged in metabolite products with defense system activation. Here, there are significant protein families, like CAMK, which are crucial for controlling the cell cycle when metabolite production is taking place.

## Transporters involved in biosynthesis of secondary pathways

Since the key role of transporters in accumulation of secondary metabolites in specific tissues is undeniable [72], in this study, 7,265 unigenes from selected modules were analyzed in the TCDB database to identify the transporters involved in the biosynthesis pathway of secondary metabolites in *R. officinalis*. Therefore, due to the distinction between the location of biosynthesis and storage of secondary metabolites in plants, identifying their transport pathways for accumulation in specific tissues or subcellular compartments has attracted more attention [73]. In most cases, the transferring of secondary metabolites is carried out intracellularly, intracellularly, or in an intra-tissue manner with the help of transporters [73]. One of the most

important families of transporters is ABC (ATP-binding cassette) [74]. ABC transporters are membrane proteins that carry metabolites in two lipid layers using energy provided by ATP hydrolysis [75]. Hence, two issues highlight the importance of ABC transporters in complementing our results: (i) since most plant secondary metabolites are used as medicines, ABC transporters are associated with drug resistance [76], and (ii) ABC transporters have characterized among the MeJA up-regulated transcripts [77].

It has been shown that ABC transporters play a role in the transport of plant secondary metabolites including alkaloids, terpenoids and phenols [78, 79]. This role has been proven in several studies through the artemisinin yield in *Artemisia annua* [80], the expression of PDR-type ABC transporter gene PgPDR3 in *Panax ginseng* [81], and the expression of a large proportion of ABC transporters in *Lycoris aurea* [77], under MeJA treatment. As well, it has demonstrated that the major facilitator superfamily (MFS) is necessary for plant tolerance against *Fusarium oxysporum* in *Populus trichocarpa* [82] and strong light in *Arabidopsis* [83], while their role has not assessed against MeJA treatment yet. In addition to these results, a large proportion of ABC transporters, which are subfamily of primary active transporters system based on the TCDB, were identified in MeJA-treated *R. officinalis* by analyzing the DEGs (Fig 3). Therefore, these results provide a significant understanding of the ABC transporters involved in the transport of plant secondary metabolites, especially in *R. officinalis*.

## Validation of hub genes correlated with secondary metabolites

The identification of hub genes involved in metabolites biosynthesis help to find an avenue for their large-scale production. Hence, visualization of the selected modules was performed to identify the hub genes that had the greatest number of connections with the other genes. *R. officinalis* may contain these hub genes that increase the number of secondary metabolites more efficiently. Some studies in the field of metabolomics coupled with RNA sequencing analysis in *Salvia sclarea* [84], *S. miltiorrhiza* [85], and *S. fruticosa* [86] have shown the effective role of CPS enzyme in the biosynthesis of abietane diterpenes, such as carnosic acid and other phenolic diterpenes. These studies showed that the accumulation of abietane diterpenes is increased by MeJA elicitor, which is in line with our results. However, there are many reports regarding the use of high concentrations of MeJA ($\geq 100$ μM) to stimulate the biosynthetic pathway of secondary metabolites [87, 88]. Therefore, our data showed that the expression level of *CPS* was increased after 12 h and then decreased after 24 h. This decrease in expression may be due to another role of CPS enzyme, i.e., gibberellin biosynthesis, to suspend plant growth and focus on the production of abietane diterpenes to enhance the plant's immune system against MeJA. In addition, it has demonstrated that MeJA can increase the expression of *PAL*, *TAT*, *C4H* and *RAS* involved in the biosynthesis pathway of rosmarinic acid [89].

Hence, in the phenylpropanoid pathway, L-phenylalanine converts to t-Cinnamic acid by PAL enzyme, followed by t-Cinnamic acid converts to 4-Coumaric acid by C4H enzyme. In addition, in the Tyrosine-derived pathway, L-tyrosine converts to 4-Hydroxyphenylpyruvic acid by TAT enzyme. Consequently, the final products of two pathways are combined by RAS enzyme and converted into rosmarinic acid through several reactions. Also, there is a positive relation between *PvTAT* expression and rosmarinic acid accumulation in *Prunella vulgaris* [90]. Same results were obtained in *Ocimum basilicum* [91] and *Agastache rugosa* [92]. However, our results showed that the expression level of *TAT*, *C4H*, and *RAS* increased over time, while the expression level of *PAL* increased after 12 h and then decreased after 24 h. It is possible that the plant was able to strengthen the defense system in the first 12 h. In this way, MYB58 and MYB63 are two TFs that positively regulate the general phenylpropanoid pathway as well as the monolignol-specific pathway. Accordingly, in this study, the expression level of

*MYB* increased over time under MeJA treatment. This strongly suggests that the biosynthetic pathway of rosmarinic acid is activated by MeJA.

The studies have demonstrated that *CIN* involves in the biosynthesis pathway of terpenes, especially 1, 8-Cineole, in *Melissa offiinalis*, [93], *Melaleuca alternifolia* [94], *Salvia officinalis* [95], and *Salvia fruticosa* [86]. By comparing to our results, the expression level of *CIN* increased over time under MeJA treatment, which was in the line with previous studies. Therefore, acquiring this information can give a deeper understanding of the genes involved in the biosynthesis of various secondary metabolites in *R. officinalis* and subsequently increase their production. Generally, the present study's findings by identifying the genes involved in the secondary metabolites biosynthesis of *R. officinalis* may open a way to produce those metabolites on a large scale.

## Conclusions

Most of *R. officinalis* applications are related to medicine and pharmacy. Consequently, breeding *R. officinalis* is aimed at increasing the biosynthesis of metabolites through metabolic engineering. On the basis of the computational systems biology analysis of the transcriptome and metabolome data, *R. officinalis* demonstrates that the co-expression modules correlate with secondary metabolite synthesis. Functional annotators and enrichment analyses have been conducted on the modules and most unigenes that are involved in metabolite biosynthesis pathways. In addition, TFs, PKs, transporters and hub genes were identified in the selected modules. These results indicate that the identified hub genes, such as *Copalyl diphosphate synthase*, *Phenylalanine ammonia lyase*, *Cineole synthase*, *Rosmarinic acid synthase*, *Tyrosine aminotransferase*, *Cinnamate 4-hydroxylase* and *MYB58*, may play an important role in *R. officinalis* metabolism. After bioinformatics analysis, qRT-PCR was conducted to confirm the results. It may be possible to use these genes in the future as candidate genes for genetic and metabolic engineering research in order to increase *R. officinalis* biosynthesis of secondary metabolites.

## Supporting information

**S1 Table. List of protein kinase families and sub families in the selected modules.**
(DOCX)

**S2 Table. List of transporter families and sub families in the selected modules.**
(XLSX)

## Acknowledgments

The authors would like to thank the Institute of Biotechnology and the Bioinformatics Research Group in the College of Agriculture.

## Author Contributions

**Conceptualization:** Ali Moghadam, Ahmad Tahmasebi.

**Data curation:** Ali Moghadam, Ahmad Tahmasebi, Mohammad Sadegh Taghizadeh.

**Formal analysis:** Eisa Foroozan, Mohammad Sadegh Taghizadeh.

**Investigation:** Ali Moghadam, Eisa Foroozan, Mohammad Sadegh Taghizadeh, Mohammad Bolhassani, Morteza Jafari.

**Methodology:** Ali Moghadam, Eisa Foroozan, Ahmad Tahmasebi, Mohammad Sadegh Taghizadeh, Mohammad Bolhassani, Morteza Jafari.

**Resources:** Ali Moghadam.

**Software:** Ali Moghadam, Eisa Foroozan, Ahmad Tahmasebi, Mohammad Sadegh Taghizadeh.

**Supervision:** Ali Moghadam.

**Validation:** Ali Moghadam, Ahmad Tahmasebi.

**Visualization:** Ali Moghadam, Ahmad Tahmasebi.

**Writing – original draft:** Eisa Foroozan, Mohammad Sadegh Taghizadeh, Mohammad Bolhassani, Morteza Jafari.

**Writing – review & editing:** Ali Moghadam, Ahmad Tahmasebi.

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
