## [Decision Letter · Decision Letter 0]

13 Oct 2022

PONE-D-22-11553

System network analysis of Rosmarinus officinalis transcriptome ─ key genes in biosynthesis of secondary metabolites

PLOS ONE

Dear Dr. Moghadam,

Thank you for submitting your manuscript to PLOS ONE. After careful consideration, we feel that it has merit but does not fully meet PLOS ONE’s publication criteria as it currently stands. Therefore, we invite you to submit a revised version of the manuscript that addresses the points raised during the review process.

We look forward to receiving your revised manuscript.

Kind regards,

Farrukh Azeem

Academic Editor

PLOS ONE

A clean copy of the edited manuscript (uploaded as the new *manuscript* file).

“The authors would like to thank the Institute of Biotechnology for supporting this research and the Bioinformatics Research Group in the College of Agriculture (Shiraz University).”

“The author(s) received no specific funding for this work”

Reviewers' comments:

Reviewer's Responses to Questions

**Comments to the Author**

1. Is the manuscript technically sound, and do the data support the conclusions?

Reviewer #1: Partly

Reviewer #2: Partly

Reviewer #3: Yes

2. Has the statistical analysis been performed appropriately and rigorously? 

Reviewer #1: I Don't Know

Reviewer #2: No

Reviewer #3: Yes

3. Have the authors made all data underlying the findings in their manuscript fully available?

Reviewer #1: No

Reviewer #2: Yes

Reviewer #3: Yes

4. Is the manuscript presented in an intelligible fashion and written in standard English?

Reviewer #1: Yes

Reviewer #2: No

Reviewer #3: No

5. Review Comments to the Author

Reviewer #1: The manuscript (research article) explores the relationship between crucial secondary metabolites and their biosynthetic genes in the rosemary. The article’s topic fits within the scope of PLOS ONE, and the argument is of interest to a broad audience. However, I have got some comments:

1. In the manuscript, the order of references is chaotic. Please check reference outputs.

2. It seems that the author is unclear about the form of the gene and protein (the name of the gene should be in italics).

3. The author used RT-PCR to confirm the function of hub genes, but only

chose three genes. It is necessary to add more genes to improve the reliability of conclusions.

4. Figures 3, 4, and 5, please, combine them.

5. Line 107: The author pointed out that ‘Many plant species treated with methyl jasmonate are able to produce compounds such as alkaloids, terpenoids, coumarins, and phenolic compounds [2]. On the other hand, the accumulation of rosmarinic acid in cell culture of methyl jasmonate-treated Menta balsamea has been proven [27].’. To our knowledge, the articles about the effect of methyl Jasmonate in Rosemary have been published. Please add relevant literature.

Yao D, Chen Y, Xu X, et al. Exploring the Effect of Methyl Jasmonate on the Expression of microRNAs Involved in Biosynthesis of Active Compounds of Rosemary Cell Suspension Cultures through RNA-Sequencing[J]. International journal of molecular sciences, 2022, 23(7): 3704.

Yao D, Zhang Z, Chen Y, et al. Transcriptome Analysis Reveals Differentially Expressed Genes That Regulate Biosynthesis of the Active Compounds with Methyl Jasmonate in Rosemary Suspension Cells[J]. Genes, 2021, 13(1): 67.

Reviewer #2: The paper needs revision, and the following questions/points should be further clarified prior to its acceptance for publication.

1. The author disregards the manuscript writing format. All citations in the text are cited in the wrong way. The citation should begin with [1]. The authors are doing the exact opposite. This reflects authors lack of expertise in scientific manuscript writing. Please revised your manuscript.

2. The references also writing in the wrong way and format. Please refer PLOS ONE format.

3. The introduction provides background information to establish the context of the study, but you should add more details about Rosmarinus spp. Benefits, uses in industry and others in other family representatives. Add more citations to show the species' significance in this study.

4. The materials and methods section needs revision and lacking many important details that make this study impossible to replicate. The arrangement of material and methods need to be improve and started with Plant material. Here are the following questions that need to be answered:

• Line 113: explain why the samples treated for 12 and 24 hours? Add the references.

• Line 113: How many leaves (g) were collected for each treatment? How many replications for each treatment? Are those biological replications?

• What data the statistical analysis used for? Please mention in the text. Please add the company and country for SAS 9.4 software (company, country). Also, for the rest software and equipments in the text.

5. The results provide interesting information but are described in a boring way. Some figures have low quality and they do not show what they should. In some cases, it is difficult to read hub gene. Figure 1 is not necessary.

• What does the colour indicate in Figure 5? Please specify in the text how the colours red, green, and white are represented in the scale and in the image.

• I suggest splitting the Figure 7. Current arrangement makes the hub genes are difficult to read.

• Figure 1 is not necessary.

6. The discussion is quite interesting, but difficult to follow due to incorrect citation structure.

7. They many grammatical errors throughout the manuscript that need to be fixed.

8. I recommend English editing by native specialists. It was difficult to read some sentences of the main text.

Reviewer #3: The manuscript by Eisa Foroozan et al. aim to reveal the key genes involved in the biosynthesis of secondary metabolites in Rosmarinus officinalis. The topic is valuable and is well within the scope of journal. But its main disadvantage is related to the unsuitable method of plant materials collection. Sometimes the text is confusing and some concepts are not clear or correct (see specific comments below). Thus, it must be substantially improved.

My concerns are the following:

1) keywords: It should be listed in alphabetical order.

2) Introduction: I don’t think the opening paragraph (line 43-47) placed here is necessary, which is not closely related to the main point of this paper. Instead, authors should directly introduce the special value of Rosmarinus officinalis in the beginning.

3) Line 49: Rosmarinic acid, carnosic acid, carnosol, and essential oils has which important medicinal value should be described more detailedly.

4) Line 54-56: Authors should simplify the statements to improve the readability and scientificity of the paper. My suggestion: Although the secondary metabolites have diverse functions in plant, including regulating growth (citing literature), interacting with the environment (citing literature), and ameliorating abiotic and biotic stress (citing literature), the structure and metabolic pathways of secondary metabolites are still unknown.

5) Plant selection and treatment: The selection of plant materials in this paper is not convincible only based on the previous literatures. Because the results drawn by other researchers cannot be same as yours due to different climate, growth conditions, and treatment methods. Therefore, I suggest authors need supplement the relevant data of the content of rosmarinic acid in Menta balsamea under methyl jasmonate treatments.

6) The figures and tables cited in the end of each sentence should be enclosed in brackets (see Line 137, 140, 141, 143 and other places).

7) Line 143: change ‘the major modules’ to ‘the three major modules’.

8) Line 148: Our results revealed that secondary metabolites have significant correlation with the co-expression modules.

9) Line 152-154: ‘While rosmarinic acid… brown4 module.’ I can’t understand what the authors mean. This sentence should be rewritten.

10) Line 157: (Table 2 & 3)

11) Line 166: replace ‘effect’ to ‘affect’.

12) Line 173: Which important family of proteins was identified in all modules except for Yellow?

13) Line 182: It should be Table S1 rather than S1 Table.

14) Line 187: This sentence should be revised as ‘The main systems are listed as follows: Channels/Pores, Electrochemical Potential-driven Transporters, Primary Active Transporters, Group Translocators, Transmembrane Electron Carriers, Accessory Factors Involved in Transport, and Incompletely Characterized Transport Systems.’

15) Authors are responsible for preparing their papers in correct English language. Some sentences are difficult to understand (e.g., Line 193-196), and the manuscript requires substantial grammatical revisions in the present form.

17) Fig. 7: The image resolution is bad. Please export the picture with the suitable format, such as jpg., tiff.

16) Discussion: In general, the discussion is too long and it should be shorten. If authors detect the content of rosmarinic acid in Menta balsamea, the discussion can be connected to the results better. Line 209-234, this part is mainly the replication of the results. Line 211: change ‘a disease’ to ‘disease’. The GO enrichment analysis and KEGG enrichment analysis need to add the relevant figures as supplementary materials.

17) Table 4: Don’t make the table on the different pages.

18) Line 547: Miss the title ‘Figure caption’.

6. PLOS authors have the option to publish the peer review history of their article (what does this mean?). If published, this will include your full peer review and any attached files.

Reviewer #1: No

Reviewer #2: No

Reviewer #3: No

---

## [Author Response · Author response to Decision Letter 0]

2 Jan 2023

To

Prof. Farrukh Azeem

Editor, Journal of PLOS ONE

Dear Prof. Farrukh Azeem

I have already submitted this manuscript with ID: PONE-D-22-11553 to your journal and after peer review, you asked me to resubmit after addressing all comments. Please find attached our revised manuscript “System network analysis of Rosmarinus officinalis transcriptome and metabolome─ key genes in biosynthesis of secondary metabolites” for publication in Plos One.

We have revised the manuscript, as requested by the reviewers, and hope that it can now be accepted for publication. Please find attached a detailed point-by-point response to the reviewer’s comments below. It is noted that all changes in the main text have conducted with track changes.

Thank you for Editorial handling and consideration of this manuscript. 

Yours sincerely, 

Ali Moghadam (corresponding author, on behalf of co-author)

Assistance Professor of Plant Biotechnology 

Reviewers’ comments

Reviewer 1:

1. In the manuscript, the order of references is chaotic. Please check reference outputs.

Response: Thank you for your help. We corrected them.

2. It seems that the author is unclear about the form of the gene and protein (the name of the gene should be in italics).

Response: You are right. We corrected them around the whole manuscript.

3. The author used RT-PCR to confirm the function of hub genes, but only chose three genes. It is necessary to add more genes to improve the reliability of conclusions.

Response: Thank you for your statement. We also analyzed four other genes, including CIN, RAS, TAT, and C4H genes, by Real-time PCR and included the results in the text.

4. Figures 3, 4, and 5, please, combine them.

Response: Thank you for your suggestion. We combined them to illustrate an image.

5. Line 107: The author pointed out that ‘Many plant species treated with methyl jasmonate are able to produce compounds such as alkaloids, terpenoids, coumarins, and phenolic compounds [2]. On the other hand, the accumulation of rosmarinic acid in cell culture of methyl jasmonate-treated Menta balsamea has been proven [27].’. To our knowledge, the articles about the effect of methyl Jasmonate in Rosemary have been published. Please add relevant literature.

Yao D, Chen Y, Xu X, et al. Exploring the Effect of Methyl Jasmonate on the Expression of microRNAs Involved in Biosynthesis of Active Compounds of Rosemary Cell Suspension Cultures through RNA-Sequencing[J]. International journal of molecular sciences, 2022, 23(7): 3704.

Yao D, Zhang Z, Chen Y, et al. Transcriptome Analysis Reveals Differentially Expressed Genes That Regulate Biosynthesis of the Active Compounds with Methyl Jasmonate in Rosemary Suspension Cells[J]. Genes, 2021, 13(1): 67.

Response: We would like to thank you for your suggestion. We added these citations to the main text.

Reviewer 2:

1. The author disregards the manuscript writing format. All citations in the text are cited in the wrong way. The citation should begin with [1]. The authors are doing the exact opposite. This reflects authors lack of expertise in scientific manuscript writing. Please revised your manuscript.

Response: We are sorry for this mistake. We corrected them around the whole manuscript.

2. The references also writing in the wrong way and format. Please refer PLOS ONE format.

Response: Thank you for your statement. We prepared them according to the PLOS ONE journal format.

3. The introduction provides background information to establish the context of the study, but you should add more details about Rosmarinus spp. Benefits, uses in industry and others in other family representatives. Add more citations to show the species' significance in this study.

Response: Thank you for your consideration and your suggestion. We updated the introduction as you wished.

4. The materials and methods section needs revision and lacking many important details that make this study impossible to replicate. The arrangement of material and methods need to be improve and started with Plant material. Here are the following questions that need to be answered:

• Line 113: explain why the samples treated for 12 and 24 hours? Add the references.

• Line 113: How many leaves (g) were collected for each treatment? How many replications for each treatment? Are those biological replications?

• What data the statistical analysis used for? Please mention in the text. Please add the company and country for SAS 9.4 software (company, country). Also, for the rest software and equipments in the text.

Response: It is a good question. Two new references were added in the text to clear this doubt. In the explanation, I should say that the results of other researchers in methyl jasmonate treatment to measure transcript changes are the reason for this.

Actually, we made powder of 15 leaves for each sample in the liquid nitrogen and used just 100 mg powder for RNA extraction for each sample. We used three biological replicates for the treatments and controls.

As we mentioned in the text, we used the In vivo data, including Real-time PCR data, for the statistical analysis. In addition, we added the company and country for SAS and GraphPad prism software in the text.

5. The results provide interesting information but are described in a boring way. Some figures have low quality and they do not show what they should. In some cases, it is difficult to read hub gene. Figure 1 is not necessary.

• What does the colour indicate in Figure 5? Please specify in the text how the colours red, green, and white are represented in the scale and in the image.

• I suggest splitting the Figure 7. Current arrangement makes the hub genes are difficult to read.

• Figure 1 is not necessary.

Response: Thank you for your valuable comments. In Fig 5, each cell shows the correlation between modules and secondary metabolites, this ratio has a range from -1 to +1. The more this ratio closer to +1, the color changes from green to red. This description added to the relevant figure caption. 

We combined the modules in Fig. 7 to avoid increasing the number of figures. Also, we combined the figures of 3, 4, and 5 to decrease the number of figures. However, we increased the resolution of images, so that the hub genes are clearly readable by zooming the image 7 (now it is image 4 in the text).

In addition, we deleted Fig. 1.

6. The discussion is quite interesting, but difficult to follow due to incorrect citation structure.

Response: Thank you for your consideration. We updated the discussion.

Reviewer 3:

1) keywords: It should be listed in alphabetical order.

Response: Thank you for your suggestion. We updated it.

2) Introduction: I don’t think the opening paragraph (line 43-47) placed here is necessary, which is not closely related to the main point of this paper. Instead, authors should directly introduce the special value of Rosmarinus officinalis in the beginning.

Response: Thank you for your suggestion. We updated the introduction as you wished.

3) Line 49: Rosmarinic acid, carnosic acid, carnosol, and essential oils has which important medicinal value should be described more detailedly.

Response: Thank you again. We added more details.

4) Line 54-56: Authors should simplify the statements to improve the readability and scientificity of the paper. My suggestion: Although the secondary metabolites have diverse functions in plant, including regulating growth (citing literature), interacting with the environment (citing literature), and ameliorating abiotic and biotic stress (citing literature), the structure and metabolic pathways of secondary metabolites are still unknown.

Response: We would like to thank you for your valuable suggestion. We added it in the main text as you wished.

5) Plant selection and treatment: The selection of plant materials in this paper is not convincible only based on the previous literatures. Because the results drawn by other researchers cannot be same as yours due to different climate, growth conditions, and treatment methods. Therefore, I suggest authors need supplement the relevant data of the content of rosmarinic acid in Menta balsamea under methyl jasmonate treatments.

Response: Thank you so much for your valuable comment. It should be noted that we currently do not have this plant cultivated and if we want to cultivate it and treat it using MeJA, it will take a lot of time. On the other hand, the species we worked with was R. officinalis. Therefore, it is better to leave it to the next research.

6) The figures and tables cited in the end of each sentence should be enclosed in brackets (see Line 137, 140, 141, 143 and other places).

Response: Thank you for your attention. We corrected them.

7) Line 143: change ‘the major modules’ to ‘the three major modules’.

Response: Thanks. We revised it.

8) Line 148: Our results revealed that secondary metabolites have significant correlation with the co-expression modules.

Response: Thank you again. We revised it.

9) Line 152-154: ‘While rosmarinic acid… brown4 module.’ I can’t understand what the authors mean. This sentence should be rewritten.

Response: Thank you for your consideration. We rewrote it.

10) Line 157: (Table 2 & 3)

Response: Thank you again. We corrected it.

11) Line 166: replace ‘effect’ to ‘affect’.

Response: Thank you. We corrected it.

12) Line 173: Which important family of proteins was identified in all modules except for Yellow?

Response: We appreciate your consideration. The analyses show that among MYB, C3H, GRAS, SBP, C2C2, HB-HD-ZIP and MADS, which identified in stated modules, the yellow module doesn’t contain MADS. In addition, we grammatically edited relevant paragraph.

13) Line 182: It should be Table S1 rather than S1 Table.

Response: Thank you. We corrected it.

14) Line 187: This sentence should be revised as ‘The main systems are listed as follows: Channels/Pores, Electrochemical Potential-driven Transporters, Primary Active Transporters, Group Translocators, Transmembrane Electron Carriers, Accessory Factors Involved in Transport, and Incompletely Characterized Transport Systems.’

Response: Thank you for your suggestion. We revised it.

17) Fig. 7: The image resolution is bad. Please export the picture with the suitable format, such as jpg., tiff.

Response: Thank you for your consideration. We enhanced the resolution of images and added to the manuscript.

16) Discussion: In general, the discussion is too long and it should be shorten. If authors detect the content of rosmarinic acid in Menta balsamea, the discussion can be connected to the results better. Line 209-234, this part is mainly the replication of the results. Line 211: change ‘a disease’ to ‘disease’. The GO enrichment analysis and KEGG enrichment analysis need to add the relevant figures as supplementary materials.

Response: The reason for the long discussion is because of the identification of several hub genes and the evaluation and discussion of the results related to them. On the other hand, we also identified TFs, transporters and protein kinases involved in the biosynthesis pathway of secondary metabolites of R. officinalis, which will naturally prolong the discussion. However, we have reduced the content of the discussion as much as possible. The output of GO and KEGG analysis was as a table containing genes and abundance, which we included it in the text.

17) Table 4: Don’t make the table on the different pages.

Response: Due to the large number of TFs, the desired table has become very long. However, we resized the table to be placed on a page. If it is possible, in the proof version, we request that it be placed as landscape.

18) Line 547: Miss the title ‘Figure caption’.

Response: Thank you for your consideration. We included it.

---

## [Editor Report · Decision Letter 1]

14 Feb 2023

System network analysis of Rosmarinus officinalis transcriptome and metabolome ─ key genes in biosynthesis of secondary metabolites

PONE-D-22-11553R1

Dear Dr. Moghadam,

We’re pleased to inform you that your manuscript has been judged scientifically suitable for publication and will be formally accepted for publication once it meets all outstanding technical requirements.

Kind regards,

Wujun Ma

Academic Editor

PLOS ONE
---

## [Editor Report · Acceptance letter]

21 Feb 2023

PONE-D-22-11553R1 

System network analysis of *Rosmarinus officinalis* transcriptome and metabolome ─ key genes in biosynthesis of secondary metabolites 

Dear Dr. Moghadam:

I'm pleased to inform you that your manuscript has been deemed suitable for publication in PLOS ONE. Congratulations! Your manuscript is now with our production department. 

Kind regards, 

on behalf of

Prof. Wujun Ma 

Academic Editor

PLOS ONE